# SEG-AGENT: IMPROVING LANGUAGE-GUIDED SEGMENTATION VIA EXPLICIT CHAIN-OF-REASONING CONSTRUCTION

## ABSTRACT

Language-guided segmentation breaks through the scope limitations of traditional semantic segmentation, enabling models to segment any target region in an image based on user instructions. Existing methods are typically two-stage frameworks: they first employ multimodal large language models (MLLMs) to understand the textual instruction and generate visual prompts from the image, and then use foundational segmentation models such as SAM to produce high-quality masks. However, due to the limited spatial grounding capability of the base models, they usually require training on large-scale datasets to achieve improved segmentation accuracy. In this paper, we propose **Seg-Agent**, a completely training-free language-guided segmentation method. By constructing an explicit reasoning chain: generation, selection, and refinement, Seg-Agent achieves performance comparable to training-based approaches. Additionally, to evaluate the generalization ability of Seg-Agent, we collect a diverse dataset covering various language-guided segmentation scenarios, named **Various-LangSeg**. Extensive experiments demonstrate the effectiveness of our proposed method. The code and dataset will be made publicly available.

## 1 INTRODUCTION

The rapid development of multimodal large language models (MLLMs) (Liu et al., 2023b; Achiam et al., 2023; Bai et al., 2025) and foundational segmentation models (Cheng et al., 2021; 2022; Kirillov et al., 2023; Ravi et al., 2024) has driven significant progress in language-guided segmentation (Ren et al., 2024a; Lai et al., 2024). Unlike traditional segmentation methods (Xie et al., 2021; Zheng et al., 2021; Liu et al., 2023a; Hao et al., 2025), which are limited to predefined categories and scenarios, language-guided segmentation models can segment any target region of interest based on textual instruction. This makes it an open and domain-unrestricted segmentation approach. As shown in Figure 1, we categorize common segmentation tasks into three types: explicit semantic segmentation (ESS), generic object segmentation (GOS), and reasoning-guided segmentation (RGS). Traditional segmentation models can typically handle only a limited subset of these scenarios. However, with the powerful understanding capabilities of MLLMs and the flexible configuration of text prompts, language-guided segmentation models are capable of addressing all three categories.

Most existing language-guided segmentation models follow a two-stage approach (Lai et al., 2024; Ren et al., 2024b; Chen et al., 2024; Liu et al., 2025). First, they use MLLMs to understand the instruction and perceive the image, generating visual prompts (typically in the form of bounding boxes or points). Then, a foundational segmentation model such as SAM is employed to produce high-quality segmentation masks based on these visual prompts. However, due to limitations such as the MLLM's relatively weak spatial perception and grounding capabilities, the visual prompts it generates directly are often of low quality (Lai et al., 2024; Yang et al., 2023). As a result, these models typically require training on large-scale datasets to improve performance.

However, these training-based methods have several notable limitations. First, it requires collecting large datasets for training. Due to the diversity of segmentation scenarios, it is difficult to fully cover all possible cases in the training data, which limits the model's generalization ability and leads to poor performance in out-of-distribution (OOD) scenarios. Second, training models requires sub-

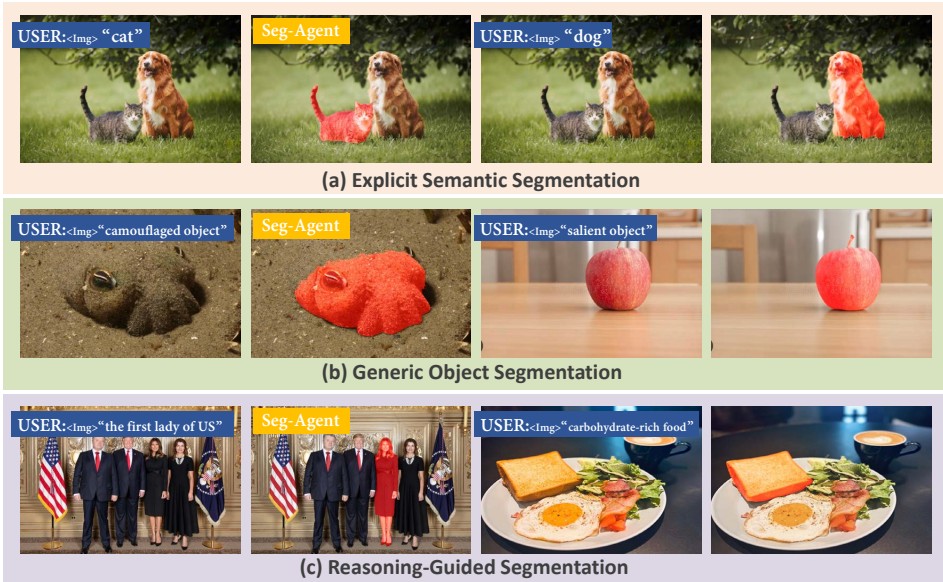

Figure 1: Given an image and a textual target, Seg-Agent can handle segmentation tasks across various scenarios: (a) Explicit Semantic Segmentation: segmenting objects with clearly defined semantics (e.g., "cat", "dog"). (b) Generic Object Segmentation: segmenting conceptually defined objects without specific categories (e.g., "camouflaged object", "salient object"). (c) Reasoning-Guided Segmentation: segmenting targets based on prompts that require commonsense or factual reasoning (e.g., "the first lady of US", "carbohydrate-rich food").

stantial computational resources, especially for MLLMs. And as newer, more powerful base models become available, training-based methods cannot be directly integrated with them. Therefore, in this paper, we propose **Seg-Agent**, a completely training-free language-guided segmentation framework. As shown in Figure 2, existing methods are all training-based, either fully trained or partially trained, which inevitably introduces the aforementioned inherent limitations. In fact, regardless of whether training is involved, the core objective is to improve the quality of the generated visual prompts and the final segmentation mask. Instead of enhancing model capabilities through training, we directly construct an explicit reasoning chain to guide the model. Specifically, we propose a three-step reasoning process: generation, selection, and refinement, through which the MLLM progressively improves the generated visual prompts, ultimately leading to better segmentation masks.

Furthermore, considering the limited scene diversity in existing language-guided segmentation datasets (Lai et al., 2024; Kazemzadeh et al., 2014; Mao et al., 2016) and to better validate the generalization capability of segmentation models, we collect a multi-scenario evaluation dataset called **Various-LangSeg**. Specifically, Various-LangSeg includes the three types of tasks illustrated in Figure 1: explicit semantic segmentation, generic object segmentation, and reasoning-guided segmentation, which collectively cover the majority of common language-guided segmentation scenarios. We evaluate the performance of Seg-Agent and several related language-guided segmentation models (Lai et al., 2024; Chen et al., 2024; Liu et al., 2025) on Various-LangSeg.

We summarize our contributions as follows:

- We propose Seg-Agent, a completely training-free framework for language-guided segmentation. By constructing an explicit reasoning chain for guidance, Seg-Agent achieves performance comparable to training-based methods.

- We have collected a comprehensive evaluation dataset named Various-LangSeg, which covers nearly all common scenarios of language-guided segmentation and effectively assesses models' generalization ability.

- Extensive experiments demonstrate the effectiveness of our proposed method and provide a low-cost, simple, and effective design paradigm for the community.

## 2 RELATED WORK

**Multimodal Large Language Models**. In recent years, MLLMs have achieved revolutionary progress in vision-language tasks. Models such as GPT-4 (Achiam et al., 2023) and Qwen-VL2.5 (Bai et al., 2025) have demonstrated outstanding capabilities in understanding multimodal content, giving them a natural advantage in tasks requiring joint image-text reasoning, thus driving the development of numerous downstream tasks (Rawles et al., 2024; Cheng et al., 2024). MLLMs have shown remarkable performance in visual question answering (Agrawal et al., 2016), image captioning (Ghandi et al., 2023), and multimodal reasoning (Zhang et al., 2024b). However, they typically lack fine-grained spatial perception and grounding abilities (Wu et al., 2024; Lai et al., 2024), which poses challenges for dense prediction tasks such as segmentation. Previous approaches mostly enhance grounding capabilities by training on task-specific datasets (Zhang et al., 2024a; Ren et al., 2024b). In contrast, our work leverages off-the-shelf, native MLLMs to generate visual prompts, but circumvents their limitations in spatial understanding through an explicit reasoning chain.

**Foundational Segmentation Models**. Foundational segmentation models such as Mask-Former (Cheng et al., 2021; 2022) and SAM (Kirillov et al., 2023; Ravi et al., 2024) are general-purpose models trained on large-scale data for universal segmentation. In particular, SAM introduces a promptable interface that enables segmentation with sparse visual cues like points or bounding boxes. These models offer strong generalization ability and high-quality mask prediction across various domains. In our framework, we use SAM2 as the backend segmentation module and focus on improving its performance by enhancing the quality of the visual prompts generated by MLLMs.

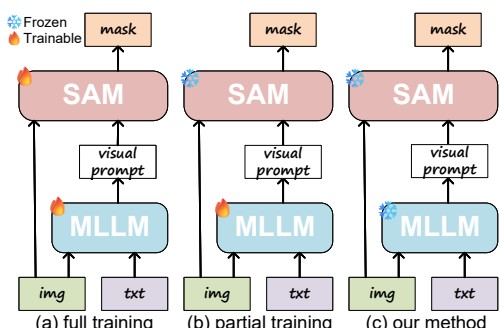

Figure 2: Comparison between the proposed Seg-Agent and existing methods. (a) Full training: both the MLLM and SAM are trained simultaneously, with representative methods including LISA (Lai et al., 2024) and Sa2VA (Yuan et al., 2025). (b) Partial training: only the MLLM is trained while SAM is kept fixed, with representative methods including SAM4MLLM (Chen et al., 2024) and Seg-Zero (Liu et al., 2025). (c) Our method is entirely training-free.

**Language-Guided Segmentation**. Early methods typically employ a text encoder (Devlin et al., 2018; Radford et al., 2021) to extract textual features for guiding the segmentation model (Ren et al., 2024a; Liu et al., 2023c; Yang et al., 2022). With the advancement of MLLMs and foundational segmentation models, most existing approaches have evolved into the two-stage framework described earlier, although they differ in subtle aspects. PixelLLM (Ren et al., 2024b) and OMG-LLaVA (Zhang et al., 2024a) use their own lightweight decoders to generate masks. However, such decoders generally underperform compared to SAM, which is pretrained on massive-scale data. As a result, current mainstream methods directly integrate SAM as the segmentation backbone. LISA (Lai et al., 2024), Sa2VA (Yuan et al., 2025) and GSVA (Xia et al., 2024) do not generate explicit visual prompts, instead, they introduce a special *<SEG>* token to compress textual information, requiring fine-tuning of SAM so that it can interpret this novel type of prompt. In contrast, SAM4MLLM (Chen et al., 2024) and Seg-Zero (Liu et al., 2025) adopt a more intuitive strategy: they keep SAM frozen and instead post-train MLLMs to generate more accurate bounding boxes or points prompts—formats that SAM can directly understand, thereby achieving improved performance. These two works are the closest to our Seg-Agent, except that we enhance the quality of generated visual prompts through a manually designed, explicit reasoning chain without any training.

## 3 VARIOUS-LANGSEG: A COMPREHENSIVE EVALUATION BENCHMARK

**Motivation** Existing language-guided segmentation datasets primarily focus on referring segmentation (Kazemzadeh et al., 2014; Mao et al., 2016) and reasoning segmentation (Lai et al., 2024). However, these datasets cover a limited spectrum of task types, which hinders the thorough evaluation of general-purpose models. To address this gap, we introduce **Various-LangSeg**, a unified and diverse benchmark designed to evaluate language-guided segmentation methods across a broad range of scenarios.

**Task Categorization** As shown in Figure 1, we categorize language-guided segmentation tasks into three representative scenarios: Explicit Semantic Segmentation, Generic Object Segmentation, and Reasoning-Guided Segmentation. These three scenarios jointly span most common language-guided segmentation tasks, enabling comprehensive evaluation of model generalization.

**Dataset Construction** Given an input image and a textual instruction, the goal is to generate the corresponding binary mask. Since our method does not require any training, Various-LangSeg is designed solely for evaluation purposes.

To construct the dataset, we proceed as follows:

1. Image Collection: We sample images and corresponding masks from existing public datasets (Lv et al., 2021; Li et al., 2014; Vicente et al., 2016; Dong et al., 2013; Lin et al., 2015).

2. Instruction Annotation: For each image-mask pair $(x_{\mathrm{img}}, y_{\mathrm{mask}})$, we manually annotate a textual instruction $x_{\mathrm{txt}}$ to form the complete triplet $(x_{\mathrm{img}}, x_{\mathrm{txt}}, y_{\mathrm{mask}})$.

Note that although the images and their corresponding masks are sourced from other datasets, we must manually analyze the relationship between each image and its corresponding mask, assign accurate textual instructions, and carefully select high-quality image-mask pairs. This process is not merely a simple sampling and assembly to form a new dataset.

For each scenario:

- **Explicit Semantic Segmentation**: 20 common object categories (e.g., cat, dog, bird) with 7 images per category, totaling 140 samples.

- **Generic Object Segmentation**: 4 binary segmentation tasks: salient object detection (SOD) (Borji et al., 2015), camouflaged object detection (COD) (Fan et al., 2020), shadow detection (SD) (Vicente et al., 2016), and image tampering detection (ITD) (Dong et al., 2013). Each task includes 16 samples. Textual input directly uses the task name (e.g., "salient object").

- **Reasoning-Guided Segmentation**: 40 samples are annotated using complex descriptions requiring implicit reasoning. These are inspired by ReasonSeg (Lai et al., 2024), such as "carbohydrate-rich food", where identification requires reasoning and domain knowledge.

In total, Various-LangSeg contains 244 evaluation samples, with 140, 64, and 40 samples in three scenarios respectively. Its scale is comparable to that of the validation set of the ReasonSeg dataset (Lai et al., 2024). We visually present the statistics of Various-LangSeg in Figure 5 in the Appendix.

**Evaluation Metrics** We follow established works (Lai et al., 2024; Liu et al., 2025) and adopt two standard metrics: **gIoU (global IoU)**: the average IoU over all samples. **cIoU (cumulative IoU)**: the ratio of the total intersection area to the total union area across the entire dataset.

## 4 METHOD

We propose Seg-Agent, a training-free framework for language-guided segmentation. Our method is a two-stage approach that first uses MLLMs to generate visual prompts and then employs a base segmentation model to produce the final mask. However, unlike previous methods that directly generate visual prompts (Chen et al., 2024; Liu et al., 2025) in a single step, we construct an explicit reasoning chain to guide the MLLM's prompt generation process, comprising generation, selection, and refinement, mimicking an iterative procedure of progressively localizing and finetuning the target boundaries. In contrast to traditional end-to-end approaches that rely on learned features, Seg-Agent explicitly constructs and updates visual prompts, guiding the segmentation model through interpretable, step-by-step interactions. The overall pipeline is illustrated in Figure 3.

### 4.1 PROBLEM FORMULATION

Let $\mathbf{X}_{\mathrm{img}} \in \mathbb{R}^{H \times W \times 3}$ be an input image and $\mathbf{X}_{\mathrm{txt}} \in \mathcal{T}$ be a natural language instruction describing the target object (e.g., "the man in red"). The goal is to predict a binary mask $\mathbf{Y}_{\mathrm{mask}} \in \{0, 1\}^{H \times W}$ that segments the described object.

Figure 3: Illustration of Seg-Agent. By constructing an explicit reasoning chain: generation, selection, and refinement, the MLLM is able to improve the quality of generated visual prompts, thereby enabling SAM to produce more accurate target segmentation masks. SoM here indicates Set-of-Mark prompt (Yang et al., 2023).

## 4.2 OVERVIEW OF SEG-AGENT PIPELINE

Seg-Agent consists of four modules:

1. **Generation Module**: Proposes diverse bounding boxes using image augmentations.
2. **Selection Module**: Selects the most appropriate box via visual comparison.
3. **Refinement Module**: Fine-tunes the selected box to better align with the object boundaries.
4. **Segmentation**: Applies a pretrained segmenter such as SAM to produce the final mask.

## 4.3 THE FORWARD PASS OF SEG-AGENT

**Generation Module** To ensure robustness across views and scales, we apply a set of augmentations $\mathcal{A} = \{a_i\}_{i=1}^{N}$ to the input image (including flipping, scaling, etc):

$$\mathbf{X}^{(i)} = a_i(\mathbf{X}_{\text{img}}), \quad i = 1, \ldots, N. \tag{1}$$

Each augmented image $\mathbf{X}^{(i)}$ is paired with $\mathbf{X}_{\text{txt}}$ and sent to an MLLM with a task-specific prompt (denoted as <generation prompt>, see Appendix) to localize the object:

$$\mathbf{B}^{(i)} = \text{MLLM}_{\text{gen}}(\mathbf{X}^{(i)}, \mathbf{X}_{\text{txt}}, \texttt{<generation prompt>}). \tag{2}$$

This yields a set of bounding box proposals (coordinate format: $[x_1, y_1, x_2, y_2]$):

$$\mathcal{B}_{\text{gen}} = \left\{ \mathbf{B}^{(i)} \right\}_{i=1}^{N}. \tag{3}$$

**Selection Module** To consolidate candidate boxes back into the original image frame, we invert the augmentations:

$$\tilde{\mathbf{B}}^{(i)} = a_i^{-1}(\mathbf{B}^{(i)}). \tag{4}$$

We perform Non-Maximum Suppression (NMS) to filter redundant boxes:

$$\mathcal{B}_{\text{sel}} = \text{NMS}\left( \left\{ \tilde{\mathbf{B}}^{(i)} \right\}, \theta_{\text{IoU}} \right). \tag{5}$$

Using a visualization strategy such as Set-of-Mark (SoM) (Yang et al., 2023), we render the candidate boxes onto the original image. This method has been shown to enhance the spatial perception and grounding capability of MLLMs (Yang et al., 2023; Rawles et al., 2024), and also allows for intuitive visualization of the spatial relationship between the bounding box and the target object, as illustrated in Figure 3. The MLLM receives the SoM-marked image, textual instruction, and a comparison prompt (denoted as <selection prompt>, see Appendix) to choose the most relevant box:

$$\mathbf{B}_{\text{sel}} = \text{MLLM}_{\text{sel}}(\text{SoM}(\mathbf{X}_{\text{img}}, \mathcal{B}_{\text{sel}}), \mathbf{X}_{\text{txt}}, \mathcal{B}_{\text{sel}}, \texttt{<selection prompt>}). \tag{6}$$

**Refinement Module** The selected box $\mathbf{B}_{\text{sel}}$ may still require fine-tuning for optimal spatial coverage. We invoke a final reasoning step using another refinement prompt (denoted as <refinement prompt>, see Appendix) to refine it:

$$\mathbf{B}_{\text{refined}} = \text{MLLM}_{\text{refine}}(\text{SoM}(\mathbf{X}_{\text{img}}, \mathbf{B}_{\text{sel}}), \mathbf{X}_{\text{txt}}, \mathbf{B}_{\text{sel}}, \texttt{<refinement prompt>}). \tag{7}$$

Through this reasoning process, the MLLM is able to carefully examine the alignment between the current bounding box and the target object, and fine-tune it based on semantic and visual context, for example, by expanding, shrinking, translating, or adjusting its boundaries to achieve more precise coverage of the target region.

The final output $\mathbf{B}_{\text{refined}}$ is a well-refined bounding box that serves as a high-quality visual prompt for the subsequent segmentation task. This module significantly improves the accuracy of boundary localization and is a key step toward achieving high-precision segmentation.

**Segmentation Module** The refined bounding box is then used as the visual prompt input to a segmentation model such as SAM:

$$\mathbf{Y}_{\text{mask}} = \text{SAM}(\mathbf{X}_{\text{img}}, \mathbf{B}_{\text{refined}}). \tag{8}$$

SAM uses the bounding box as a spatial prompt to precisely identify and segment the described target region in the image, producing a high-quality, pixel-level binary mask $\mathbf{Y}_{\text{mask}}$, where 1 indicates the target region and 0 indicates the background.

This step completes the final transformation from a language instruction to an accurate segmentation, serving as the last stage of the entire Seg-Agent framework. Thanks to the progressive refinement of visual prompts in the previous three stages, SAM receives more accurate guidance, thereby significantly improving the final segmentation accuracy.

### 4.4 WHY SEG-AGENT MATTERS

Seg-Agent decomposes language-guided segmentation into interpretable sub-tasks, enabling robust performance across diverse conditions without any task-specific training. By explicitly engaging in step-wise reasoning, Seg-Agent avoids common failure modes of end-to-end systems and provides traceable decision-making, all while fully leveraging the generalization power of MLLMs.

Unlike previous works relying on parameter updates, Seg-Agent operates in a zero-shot and training-free setting, relying solely on step-wise reasoning within MLLMs. This has several advantages:

- **Generalization:** Augmentation-enriched proposals increase robustness across unseen distributions. We perform no post-training on MLLMs, thus avoiding any potential negative impacts on their performance.

- **Interpretability:** Each reasoning step is explicit and traceable, enabling transparent debugging and user intervention.

- **Modularity:** Seg-Agent can be instantly adapted to newer and stronger MLLMs or segmentation models without retraining.

## 5 EXPERIMENTS

### 5.1 EXPERIMENTAL SETTING

**Implementation Details.** We employ QwenVL-2.5 (Bai et al., 2025) as the base MLLM for generating visual prompt, the generation module, selection module, and refinement module are all built upon it, which can be deployed locally or accessed via API services. Additionally, we use SAM2-Large (Ravi et al., 2024) to generate segmentation masks. Seg-Agent operates in a training-free manner, and the entire inference process can be completed on a single NVIDIA RTX 4090 GPU with 24 GB of memory. For the NMS step, we set the IoU threshold to 0.8. During inference, the prompt for Seg-Agent is pre-defined (see Appendix),

Table 1: Referring segmentation results. Methods marked with "*" are traditional approaches, while the other methods are based on MLLMs. We compare cIoU in this table. Best results are in **bold**.

| Method | RefCOCO testA | RefCOCO+ testA | RefCOCOg test |
|---|---|---|---|
| training-based methods | | | |
| CRIS* | 73.2 | 68.1 | 60.4 |
| LAVT* | 75.8 | 68.4 | 62.1 |
| ReLA* | 76.5 | 71.0 | 66.0 |
| LISA-7B | 76.5 | 67.4 | 68.5 |
| PixelLM-7B | 76.5 | 71.7 | 70.5 |
| PerceptionGPT-7B | 78.6 | 73.9 | 71.7 |
| Seg-Zero-3B | 79.3 | 73.7 | 71.5 |
| Seg-Zero-7B | **80.3** | **76.2** | **72.6** |
| training-free methods | | | |
| Qwen2.5-VL-3B + SAM2-L | 75.9 | 71.5 | 70.1 |
| Qwen2.5-VL-7B + SAM2-L | 77.8 | 73.5 | 71.2 |
| Seg-Agent-3B (Ours) | 79.0 | 73.2 | 71.4 |
| Seg-Agent-7B (Ours) | **79.9** | **76.0** | **72.2** |

requiring only the input of the target object. For other methods, we set the text input according to the templates provided in their works.

**Datasets.** Since our method does not require training, we only select benchmark datasets to evaluate model performance. Following prior related work (Lai et al., 2024; Chen et al., 2024; Liu et al., 2025), we adopt three datasets for the referring segmentation task: refCOCO, refCOCO+ (Kazemzadeh et al., 2014), and refCOCOg (Mao et al., 2016). These datasets involve simple textual descriptions such as "the man wearing white clothes", belonging to the explicit semantic segmentation scenario. We also include the ReasonSeg (Lai et al., 2024) dataset for the reasoning segmentation task, which falls under the reasoning-guided segmentation scenario and contains referring expressions that require reasoning, such as "the food with the most Vitamin C". Finally, we introduce the Various-LangSeg dataset proposed in this paper, which covers three common scenarios in language-guided segmentation and effectively evaluates the model's generalization and versatility.

**Baseline Methods.** We compare Seg-Agent with two groups of methods: (1) Training-based methods, including traditional methods that do not use MLLMs such as grounded-sam (Ren et al., 2024a), ReLA (Liu et al., 2023a) etc., and MLLM-based methods such as LISA (Lai et al., 2024), SAM4MLLM (Chen et al., 2024), etc.; (2) Training-free methods, we mainly adopt the Qwen2.5-VL + SAM2 (Bai et al., 2025; Ravi et al., 2024) baseline method introduced in Seg-Zero (Liu et al., 2025), with related settings and prompts kept consistent with that paper, i.e., letting MLLMs directly output bounding box coordinates in a single-step method to prompt SAM, rather than having an explicit thinking process like Seg-Agent.

**Evaluation Metrics.** We follow prior work (Lai et al., 2024; Liu et al., 2025) in adopting two evaluation metrics: gIoU and cIoU, which have been described before. Since cIoU tends to be heavily biased toward large objects and exhibits high variability, gIoU is generally preferred as the primary metric.

Table 2: Reasoning segmentation results. "-" indicates the results are not available.

| Method | ReasonSeg | | | |
|---|---|---|---|---|
| | Val | | Test | |
| | gIoU | cIoU | gIoU | cIoU |
| training-based methods | | | | |
| X-Decoder* | 22.6 | 17.9 | 21.7 | 16.3 |
| SEEM* | 25.5 | 21.2 | 24.3 | 18.7 |
| ReLA* | 22.4 | 19.9 | 21.3 | 22.0 |
| OVSeg* | 28.5 | 18.6 | 26.1 | 20.8 |
| Grounded-SAM* | 26.0 | 14.5 | 21.3 | 16.4 |
| LISA-7B-LLaVA1.5 | 53.6 | 52.3 | 48.7 | 48.8 |
| LISA-13B-LLaVA1.5 | 57.7 | 60.3 | 53.8 | 50.8 |
| SAM4MLLM | 46.7 | 48.1 | - | - |
| Seg-Zero-3B | 58.2 | 53.1 | 56.1 | 48.6 |
| Seg-Zero-7B | **62.6** | **62.0** | **57.5** | **52.0** |
| training-free methods | | | | |
| Qwen2.5VL-3B+SAM2-L | 53.6 | 44.0 | 47.9 | 37.8 |
| Qwen2.5VL-7B+SAM2-L | 57.6 | 48.3 | 50.1 | 41.2 |
| Seg-Agent-3B (Ours) | 57.8 | 56.1 | 55.5 | 49.8 |
| Seg-Agent-7B (Ours) | **61.7** | **61.2** | **57.6** | **51.8** |

Table 3: Results on Various-LangSeg. We report the performance across three scenarios and the overall performance.

| Method | Various-LangSeg | | | | | | | |
|---|---|---|---|---|---|---|---|---|
| | Explicit Semantic | | Generic Object | | Reasoning-Guided | | Overall | |
| | gIoU | cIoU | gIoU | cIoU | gIoU | cIoU | gIoU | cIoU |
| training-based methods | | | | | | | | |
| ReLA* | 76.8 | 77.7 | 20.1 | 22.1 | 25.2 | 21.2 | 53.4 | 53.8 |
| OVSeg* | 77.1 | 76.0 | 23.2 | 23.0 | 25.0 | 22.1 | 54.4 | 53.2 |
| LISA-7B | 81.9 | 83.1 | 32.4 | 30.8 | 46.8 | 36.7 | 63.2 | 64.2 |
| LISA-13B | **82.8** | **83.9** | 35.3 | 40.3 | 54.4 | 47.9 | 65.8 | 67.9 |
| PixelLLM-7B | 81.5 | 83.5 | 31.2 | 31.5 | 45.2 | 40.1 | 62.3 | 62.7 |
| SAM4MLLM | 82.1 | 83.5 | 32.0 | 31.8 | 45.2 | 37.1 | 62.9 | 62.3 |
| Seg-Zero-7B | 81.8 | 81.0 | **41.4** | **43.5** | **74.5** | **67.1** | **70.0** | **68.9** |
| training-free methods | | | | | | | | |
| Qwen2.5-VL-3B + SAM2-L | 80.1 | 77.4 | 33.3 | 33.7 | 61.0 | 49.5 | 64.7 | 61.3 |
| Qwen2.5-VL-7B + SAM2-L | 80.8 | 79.7 | 39.5 | 38.3 | 70.1 | 58.9 | 68.2 | 65.4 |
| Seg-Agent-3B (Ours) | 82.3 | 81.5 | 40.8 | 31.5 | 66.1 | 62.6 | 68.8 | 60.9 |
| Seg-Agent-7B (Ours) | **83.0** | **83.7** | 41.0 | 42.1 | **75.2** | **66.7** | **70.6** | **68.5** |

## 5.2 COMPARISON WITH OTHER METHODS

In this subsection, we conduct a comparative analysis of the performance between Seg-Agent and several most relevant methods. We compare CRIS (Wang et al., 2022), LAVT (Yang et al., 2022),

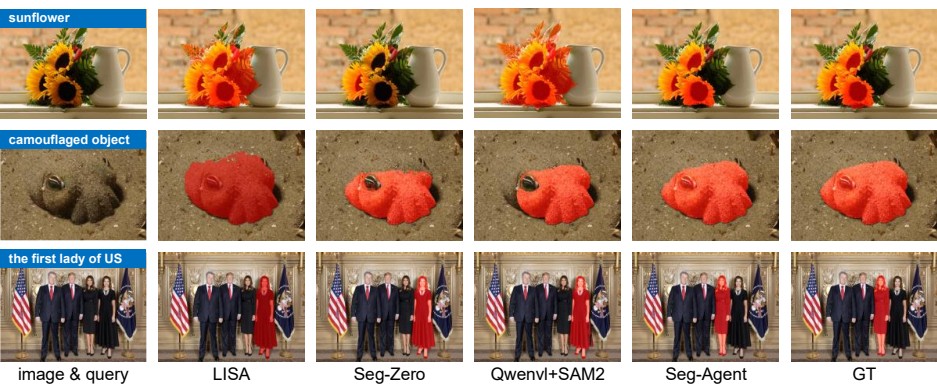

Figure 4: Visual comparison between the proposed Seg-Agent and existing related methods. We show three common scenarios of language-guided segmentation here.

OVSeg (Liang et al., 2023), X-Decoder (Zou et al., 2022), SEEM (Zou et al., 2023), ReLA (Liu et al., 2023a), LISA (Lai et al., 2024), PixelLLM (Ren et al., 2024b), PerceptionGPT (Pi et al., 2024), SAM4MLLM (Chen et al., 2024) and Seg-Zero (Liu et al., 2025).

**Referring Segmentation.** As shown in Table 1, we evaluate Seg-Agent and other related methods on the test sets of refCOCO, refCOCO+, and refCOCOg. As mentioned earlier, the target objects in these datasets are described using simple and direct text descriptions, which falls under the explicit semantic segmentation scenario. Due to the relatively simple nature of this task, all methods achieve good performance, and there is little difference between traditional methods and MLLM-based methods. Notably, the training-free baseline method Qwen2.5VL+SAM2-L also achieves favorable results, while Seg-Agent provides a certain improvement over this baseline and achieves performance comparable to the SOTA training-based method Seg-Zero.

**Reasoning Segmentation.** As shown in Table 2, we evaluate Seg-Agent and other related methods on the validation and test sets of ReasonSeg (Lai et al., 2024). As previously mentioned, the target objects in this dataset are described using reasoning-based textual expressions, categorized into long queries and short queries, falling under the reasoning-guided segmentation scenario. Compared to explicit semantic segmentation, this task requires reasoning to first identify the target object, making it significantly more challenging. As can be seen from the table, traditional methods that do not employ MLLMs perform poorly in this scenario, as their text encoders are typically better at extracting textual features but lack reasoning capabilities. In contrast, MLLM-based methods achieve substantial performance improvements. Moreover, the training-free baseline method Qwen2.5VL+SAM2-L also achieves favorable performance, while Seg-Zero achieves significant gains over this baseline through GRPO (Shao et al., 2024) training. Similarly, our Seg-Agent achieves notable improvement over the same baseline and attains performance comparable to the SOTA training-based method Seg-Zero, surpassing earlier approaches such as LISA.

**Various-LangSeg.** As shown in Table 3, we evaluate Seg-Agent and related methods on Various-LangSeg. It can be observed that traditional methods perform poorly on this dataset, achieving acceptable results only in the explicit semantic segmentation scenario, and even then, they are outperformed by MLLM-based approaches. Their performance is also weak in the general object segmentation scenario and particularly poor in the reasoning-guided segmentation scenario. In contrast, MLLM-based methods are applicable across all three scenarios. However, due to the lack of training on relevant data and the absence of an explicit reasoning process, early methods such as LISA perform relatively poorly on the general object segmentation task, underperforming compared to their results on the other two tasks. Seg-Zero achieves the best performance among training-based methods, especially in the reasoning-guided scenario, benefiting from task-specific training. For training-free methods, Seg-Agent consistently improves upon the Qwen2.5-VL+SAM2-L baseline in all three scenarios. Notably, Seg-Agent-7B achieves the best overall performance among training-free methods, and even outperforms many training-based approaches, highlighting its strong generalization and reasoning ability without task-specific training.

**Visual Comparison.** Figure 4 presents a visual comparison of Seg-Agent with several of the most relevant methods, including LISA, Seg-Zero, and Qwen2.5-VL-7B+SAM2-L. In the first row (explicit semantic segmentation scenario), the target object is a sunflower, other methods either over-

segment or under-segment the object, while only Seg-Agent produces an accurate and precise segmentation. In the second row (general object segmentation scenario), other methods fail to fully capture the main structure or produce overly coarse boundaries, whereas Seg-Agent successfully preserves the overall structure without edge blurring. In the final row (reasoning-guided segmentation scenario), only Seg-Agent correctly identifies the first lady of US (second from the right), while all other methods select incorrect objects. These visual results demonstrate the strong generalization capability and segmentation accuracy of Seg-Agent. We present more visualization results in Figure 6 in the Appendix.

## 5.3 Ablation Study

**Effectiveness of Each Module**. As shown in Table 4, the performance when combining GM with SM or RM is better than that of the baseline method. Only when all three modules are used together does the performance reach its maximum, demonstrating the effectiveness of the explicitly constructed reasoning chain in our approach. As can be observed, compared to the baseline method, the combination of different modules yields the largest performance gain in the reasoning-guided segmentation scenario, likely because this task inherently requires a reasoning process. In contrast, explicit semantic segmentation is relatively simple, and the baseline method already achieves strong performance, resulting in only marginal improvements. From the metrics, the generic object segmentation scenario appears to be the most challenging; thus, a noticeable performance improvement is achieved only when all three modules are used together. Notably, the configuration used here is consistently Qwen2.5-VL-7B + SAM2-L. In the first row, none of the three modules are employed, corresponding to the baseline method mentioned earlier with no improvements applied. When GM and RM are used together, only one candidate box needs to be generated.

Table 4: Ablation study on each module. GM, SM and RM denote generation module, selection module and refinement module, respectively. We compare gIoU here.

| GM | SM | RM | Various-LangSeg | | | |
|---|---|---|---|---|---|---|
| | | | ESS | GOS | RGS | Overall |
| ✗ | ✗ | ✗ | 80.8 | 39.5 | 70.1 | 68.2 |
| ✓ | ✓ | ✗ | 81.2 | 39.8 | 72.0 | 68.8 |
| ✓ | ✗ | ✓ | 81.5 | 40.0 | 71.8 | 69.0 |
| ✓ | ✓ | ✓ | **83.0** | **41.0** | **75.2** | **70.6** |

**Generalization across Base Models**. We conduct this experiment to demonstrate that Seg-Agent can be directly adapted to different base models. As shown in Table 5, under the same configuration, Seg-Agent, which employs an explicit reasoning chain, achieves performance improvements compared to direct inference. Moreover, it can be observed that Seg-Agent is compatible with different MLLMs (Bai et al., 2025; Zhu et al., 2025) and segmentation models (Kirillov et al., 2023; Ravi et al., 2024), and the stronger the base model, the better Seg-Agent performs. This highlights the advantage of our proposed training-free approach: as newer and more powerful base models emerge, Seg-Agent can directly integrate with them to achieve improved segmentation performance, which is a significant advantage over training-based methods.

Table 5: Ablation study on base models. a, b, and c represent different combinations of MLLMs and base segmentation models, directly using a single-step reasoning approach. Seg-Agent (x) denotes using the configuration described in x to replace the corresponding part of Seg-Agent. We compare gIoU in this table.

| Setting | Various-LangSeg | | | |
|---|---|---|---|---|
| | ESS | GOS | RGS | Overall |
| a: InternVL3-8B + SAM2-L | 79.2 | 35.3 | 67.1 | 65.7 |
| b: Qwen2.5-VL-7B + SAM-L | 80.1 | 37.5 | 69.9 | 67.3 |
| c: Qwen2.5-VL-7B + SAM2-L | 80.8 | 39.5 | 70.1 | 68.2 |
| Seg-Agent (a) | 79.8 | 36.3 | 67.5 | 66.4 |
| Seg-Agent (b) | 81.1 | 39.5 | 71.2 | 68.6 |
| Seg-Agent (c) | 83.0 | 41.0 | 75.2 | 70.6 |

## 6 Conclusion

In this paper, we propose Seg-Agent, a completely training-free language-guided segmentation model. By constructing explicit reasoning chains of generation, selection, and refinement to guide the model in generating more accurate visual prompts, Seg-Agent achieves segmentation performance comparable to training-based methods. Additionally, we construct the Various-LangSeg dataset containing multiple scenarios, which can comprehensively evaluate the generalization capability of language-guided segmentation models. Extensive experiments demonstrate the effectiveness of our approach. We hope that our simple yet effective method can provide rich inspiration to the community.

## ETHICS STATEMENT

This work adheres to the ICLR Code of Ethics. This study involved no human subjects or animal experimentation. All datasets used were sourced in compliance with relevant usage guidelines, ensuring no violation of privacy. We have taken care to avoid any biases or discriminatory outcomes in our research process. No personally identifiable information was used, and no experiments were conducted that could raise privacy or security concerns. We are committed to maintaining transparency and integrity throughout the research process.

## REPRODUCIBILITY STATEMENT

To facilitate the reproduction of the Seg-Agent proposed in this paper, we provide implementation details in Section B and an initial version of the code in the supplementary materials. Additionally, we include the JSON files describing the Various-LangSeg dataset in the supplementary materials, and we will publicly release the full code and dataset in the near future.

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

## A   The Use of Large Language Models

LLMs were used only during the writing phase, including for polishing the text and providing suggestions to improve the paper's figures.

## B   More Details about Seg-Agent

In this section, we present some implementation details of Seg-Agent. We also provide a basic version of the code in the supplementary materials, and the complete code will be released shortly.

### B.1   Generation Module

In our experiments, we observed that the object identified in a single inference pass can be entirely incorrect. Inspired by object segmentation approaches that generate multiple candidate boxes, we found that if multiple candidate boxes are produced, it becomes easier for the LLM to select the one that correctly localizes the target object. To this end, we adopted a multi-view input strategy to encourage the generation of multiple candidate boxes. Specifically, we employed common image augmentation techniques from computer vision, including flipping (horizontal and vertical) and scaling (zooming in by 2× or zooming out to 0.5×). In each case, we primarily selected two such augmented views along with the original image, resulting in three candidate boxes.

The <generation prompt> we used is as follows:

```
def build_generation_prompt(query):
    template = """Locate "{query}", report the bboxes
    coordinates in JSON format."""
    return template.format(query=query)
```

Here, the query refers to the textual description of the target object we wish to segment. The output coordinates are in the format $[x_1, y_1, x_2, y_2]$, representing the pixel coordinates of the top-left and bottom-right corners of the bounding box, e.g., `[100, 100, 200, 200]`.

### B.2   Selection Module

The Set-of-Marks (SoM) (Yang et al., 2023) used here has been shown to effectively enhance MLLMs' perception of specific objects in images and is widely employed in tasks such as screen-

shot grounding (Rawles et al., 2024). The approach involves overlaying visual bounding boxes on the original image according to the box coordinates, explicitly illustrating the spatial relationship between the target object and the generated box to help the MLLM evaluate the quality of candidate boxes. Different candidate boxes are distinguished by distinct colors, and the MLLM is provided with corresponding prompts to guide its selection.

The <selection prompt> we used is as follows:

```
def build_select_prompt(query, coords):
    prompt = f'Please analyze the image provided
    below and determine which of the bounding boxes
    better captures the "{query}".\n'

    prompt += """
Your task is to:
1. Identify which bounding box more
accurately includes the entire target object.
2. Provide a brief explanation for your choice.

Note: The format of the bounding box is
[x_min, y_min, x_max, y_max], representing the
top-left and bottom-right coordinates.

The coordinates for each bounding box are as follows:
"""
    d = 0
    colors = ['red', 'green', 'blue', 'yellow']

    for a in coords:
        prompt += f'- **Bbox {d+1} ({colors[d]})**: {a}\n'
        d += 1
    prompt += """
Return your answer in the following format:

Best Box: <Box Number>
Reasoning: <Explanation>"""
    return prompt
```

Here, "coords" refers to the list of candidate bounding boxes.

### B.3 REFINEMENT MODULE

This step mimics the human annotation process, where candidate bounding boxes are further refined to better enclose the target object.

The <refinement prompt> we used is as follows:

```
def build_optimize_prompt(query, current_box):
    prompt = f"""Please analyze the image provided below
    and evaluate whether the current bounding box
    accurately captures the "{query}".\n"""
    prompt += f"The current bounding box
    coordinates are: {current_box},
    where:\n"
    prompt += "- `x_min` = {:.2f}
    (left edge)\n".format(current_box[0])
    prompt += "- `y_min` = {:.2f}
    (top edge)\n".format(current_box[1])
    prompt += "- `x_max` = {:.2f}
    (right edge)\n".format(current_box[2])
```

```
        prompt += "- `y_max` = {:.2f}
        (bottom edge)\n".format(current_box[3])

        prompt += """

Your task is to:
1. Assess whether the current bounding
box adequately includes the entire target object.
2. If the current box does not perfectly
capture the target object or leaves unnecessary
margins, suggest an optimized bounding box
with improved coordinates.

Note: The current bounding box may
not be accurate. Please carefully
analyze the image and improve the
coordinates if necessary.

Return your response in the following format:

Current Box: [x_min, y_min, x_max, y_max]
Optimized Box:
[x_min_optimized, y_min_optimized,
x_max_optimized, y_max_optimized]
Reasoning: <Explanation of why the optimization was made>
"""
    return prompt
```

### B.4  Segmentation Module

We primarily use SAM2-L (Ravi et al., 2024) to generate binary masks, which only requires an input image and corresponding visual prompts. In our case, the visual prompts are the pixel coordinates of bounding boxes ($[x_1, y_1, x_2, y_2]$) output by the MLLM in the previous step.

## C  More Information about Various-LangSeg

We provide an overview JSON file of Various-LangSeg in the supplementary materials, which contains detailed information about the entire dataset. Due to file size limitations, we did not upload the images themselves; we will release the full dataset and evaluation code upon paper acceptance.

All images were selected from external publicly available datasets, including NC4K (Lv et al., 2021), PASCAL-S (Li et al., 2014), SBU (Vicente et al., 2016), CASIA (Dong et al., 2013), and COCO (Lin et al., 2015), from which we collected both images and their corresponding masks, and manually constructed the textual prompts. Specifically, the explicit semantic segmentation scenario includes 20 subcategories, such as cat, dog, and other objects with clear semantic meanings, with 7 samples per category, resulting in a total of 140 samples. This task is relatively simple. The reasoning-guided segmentation scenario contains 40 samples, for which we designed textual prompts requiring reasoning based on the spatial and contextual relationships between the masks and the original images. This task has moderate difficulty. Finally, the generic object segmentation scenario includes four popular binary segmentation tasks: camouflaged object detection (COD) (Fan et al., 2020), salient object detection (SOD) (Borji et al., 2015), shadow detection (SD) (Vicente et al., 2016), and image tampering detection (ITD) (Dong et al., 2013). Each subtask contains 16 samples, totaling 64 samples. This task is relatively difficult. The entire dataset comprises 244 samples in total.

Since all segmentation tasks can essentially be guided by language, Various-LangSeg certainly cannot cover all possible scenarios. However, compared to existing datasets (Lai et al., 2024; Kazemzadeh et al., 2014; Mao et al., 2016), Various-LangSeg is more comprehensive and better suited for evaluating the generalization ability of language-guided segmentation models. We also plan to extend it in the future.

Our main effort involved selecting high-quality images and their corresponding masks from existing datasets and manually crafting appropriate textual descriptions based on a careful understanding of the correspondence between each image and its mask. Regarding the dataset scale, since our goal was to establish an evaluation benchmark rather than a large-scale training set, the size of our dataset is relatively modest—comparable to the validation set of ReasonSeg (Lai et al., 2024). We plan to further expand this dataset in the future by incorporating additional segmentation scenarios.

Figure 5 shows the detailed statistics of the Various-LangSeg dataset.

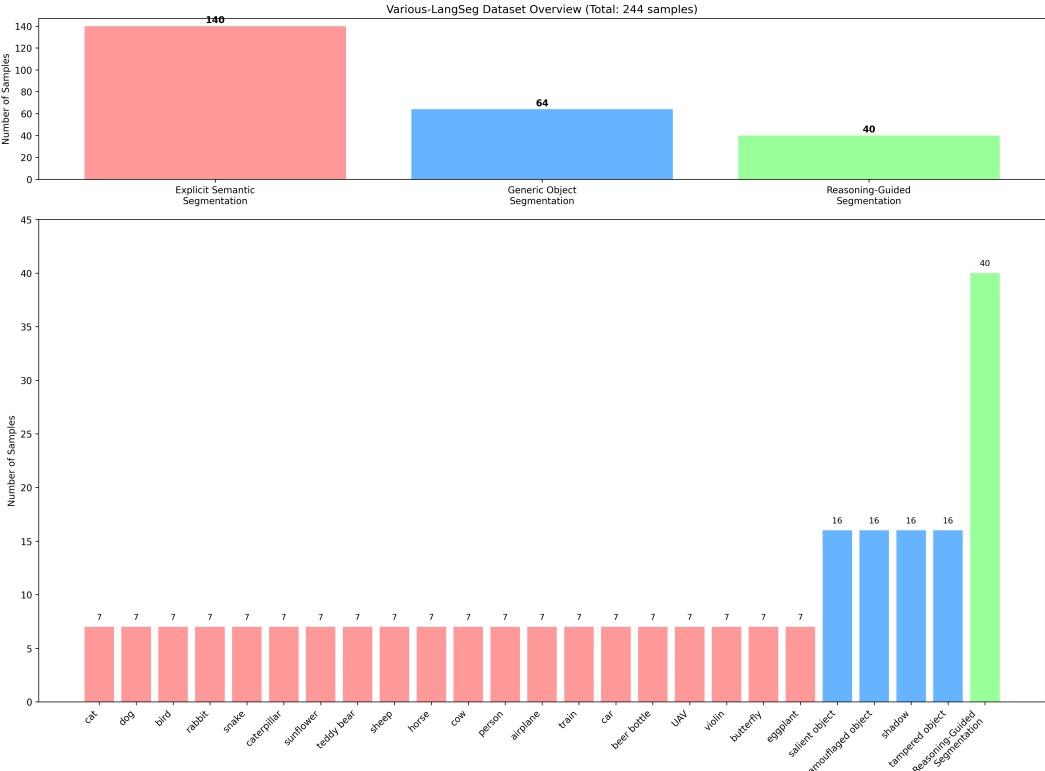

Figure 5: Overview of the Various-LangSeg dataset, comprising 244 evaluation samples across three segmentation scenarios. (Top) High-level distribution: Explicit Semantic Segmentation (140 samples), Generic Object Segmentation (64 samples), and Reasoning-Guided Segmentation (40 samples). (Bottom) Detailed breakdown: 20 explicit object categories (7 samples each), 4 generic tasks: salient object, camouflaged object, shadow, and tampered object detection (16 samples each), and the reasoning-guided category (40 samples).

## D  MORE VISUALIZATION RESULTS

We provide more visualization results in Figure 6. As can be seen, Seg-Agent is capable of handling inputs in various forms. It supports multilingual inputs, with a primary demonstration of Chinese, English, and their mixed usage. Furthermore, we present a variety of image types, including images from web news, datasets, screenshots, cartoons, photographs taken by cameras, and AI-generated images. The examples also cover the three types of language prompts introduced in the main text: explicit semantic segmentation, generic object segmentation, and reasoning-guided segmentation.

These high-quality segmentation results demonstrate the strong generalization capability and broad applicability of Seg-Agent. We have also released preliminary inference code in the supplementary materials, and welcome readers to try it out.

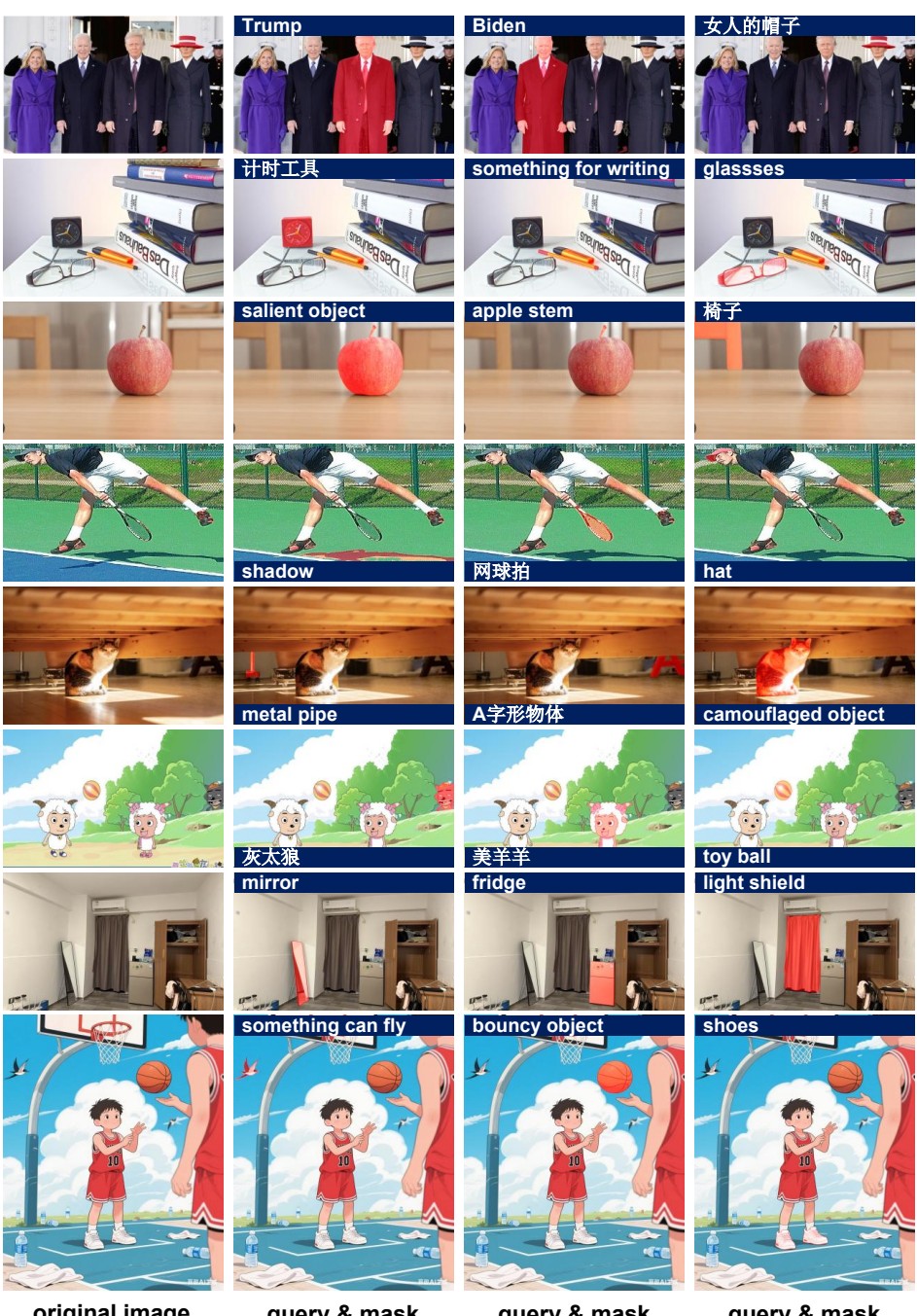

Figure 6: More visualization results of Seg-Agent. Seg-Agent can handle language inputs in various forms, including both Chinese and English. It is also capable of processing different types of images, such as real-world photos, captured photographs, cartoon images, and AI-generated images. These images demonstrate Seg-Agent's strong generalization ability and its broad range of application scenarios. Please zoom in for a better view.

## E INFERENCE PROCESS

The inference process of Seg-Agent is fully transparent and can be observed in real time. As shown in Figure 7, we present a visualization of the complete reasoning pipeline. Given an input image and a target text prompt "shadow", the generation module first produces multiple candidate bounding boxes through image augmentation and prompt-guided generation, which are then visualized on the



Figure 7: The inference process of Seg-Agent. The current target object is "shadow". GM, SM and RM indicate generation module, selection module and refinement module, respectively. Please zoom in for a better view.

original image in the form of Set-of-Mark (SoM) (Yang et al., 2023). Next, the selection module chooses the most appropriate box based on the SoM-formatted image and a selection prompt. Finally, the refinement module further improves the selected box using a refinement prompt and the SoM input. The final high-quality segmentation mask is then generated by the segmentation model using the refined bounding box as a visual prompt. The entire process is fully open and transparent. By carefully designing this reasoning pipeline, Seg-Agent effectively enhances the quality of the generated masks.

Compared to directly generating visual prompts in a single step using MLLMs, we improve the quality of visual prompts through a manually designed reasoning chain. In contrast, training-based approaches (Lai et al., 2024; Liu et al., 2025) enhance output quality by training the model to adjust its weights. While the methodologies differ, the underlying goal is essentially the same: to produce more accurate and reliable visual prompts.

## F  ANALYSIS AND DISCUSSION

The experimental results across multiple benchmarks demonstrate the effectiveness, generalizability, and flexibility of Seg-Agent. Compared with both training-based and training-free baselines, Seg-Agent consistently achieves competitive or superior performance, particularly in more complex scenarios such as reasoning-guided segmentation and mixed-scene generalization (e.g., Various-LangSeg). This validates the strength of our explicit multi-stage reasoning design, which encourages step-by-step visual grounding rather than relying on a single forward pass.

Several key observations emerge from the experiments:

- **Progressive reasoning improves localization.** Our ablation studies confirm that adding selection and refinement steps leads to significant improvements over single-step generation. This highlights the importance of decomposing the task into interpretable subtasks, especially for ambiguous or complex queries.
- **Seg-Agent is robust across tasks and data distributions.** The model performs well not only on traditional referring segmentation datasets but also on reasoning-intensive and multi-domain scenarios. This suggests that our prompt-based design allows the MLLM to adapt flexibly without task-specific training.
- **Zero-shot and modular design is practically valuable.** Unlike many training-based methods, Seg-Agent requires no fine-tuning and can easily integrate with newer MLLMs or segmentation models. This makes it a highly deployable and maintainable system in real-world applications.

## G  LIMITATIONS AND FUTURE WORK

Despite these advantages, Seg-Agent still has several limitations. First, its multi-stage reasoning process requires multiple calls to MLLMs, involving three steps: generation, selection, and refinement, which may result in higher latency compared to single-step methods. Second, the model's performance heavily depends on the quality of prompt engineering and the inherent capabilities of the MLLM itself; even with a well-designed reasoning pipeline, suboptimal results may arise if the underlying MLLM lacks sufficient reasoning or perception ability. Furthermore, the model is prone

to failure in challenging cases, such as when the target object is small or visually ambiguous, when multiple objects are present, or when the language description is vague or indirect. Of course, these issues are common challenges faced by all language-guided segmentation models. Finally, since we do not fine-tune or modify the segmentation model, performance bottlenecks in the segmentation model itself can lead to failures. For example, as shown in Figure 7, the bounding box fully encloses the shadow, yet the segmentation model fails to fully capture it.

In future work, we plan to explore more efficient prompting strategies to reduce computation while retaining accuracy, and investigate adaptive reasoning depth based on the complexity of the input query. Incorporating visual feedback loops or confidence-based control could further enhance robustness. Finally, expanding Seg-Agent to support multi-turn interaction or conversational segmentation is a promising direction for broader applicability.

