# OpenReview forum: "Seg-Agent: Improving Language-Guided Segmentation via Explicit Chain-of-Reasoning Construction"
_ICLR.cc/2026/Conference — ICLR 2026 Conference Withdrawn Submission_

### Official Review · Reviewer_3bhk · 2025-10-17

**Soundness:** 3
**Presentation:** 3
**Contribution:** 3
**Rating:** 6
**Confidence:** 5

**Summary:**

This paper proposes Seg-Agent, a training-free framework for language-guided segmentation that performs segmentation based on user instructions without additional training. The method introduces a three-stage reasoning process, generation, selection, and refinement,  to achieve performance comparable to training-based models. Additionally, the authors present Various-LangSeg, a new dataset designed to evaluate generalization across diverse segmentation scenarios.

**Strengths:**

- The paper presents a simple yet effective training-free method for language-guided segmentation.


- By introducing an explicit reasoning chain (generation, selection, and refinement),  the approach successfully decomposes a single task into multiple sub-tasks, leading to improved visual prompt quality.


- Seg-Agent achieves performance comparable to training-based methods, demonstrating strong generalization despite the absence of fine-tuning.


- The authors also propose a new evaluation benchmark, Various-LangSeg, which is built from diverse image sources and thus covers a wide range of data distributions and task types.

**Weaknesses:**

- It would be beneficial to include additional recent papers (ICLR-published works [1] and [2]) in the Language-Guided Segmentation of Related Work to provide a more comprehensive literature context.


- The model’s performance appears to be highly dependent on the pre-trained knowledge of the MLLM modules used within the framework.


- It remains unclear whether the method can effectively handle multi-target objects or part-level segmentation tasks.


- When the input query refers to a non-existent target in the image, the system seems unable to reject such cases properly. Based on the prompts shown in the appendix, the framework always predicts bounding box coordinates, revealing a limitation in rejecting absent targets.

[1] MMR: A Large-scale Benchmark Dataset for Multi-target and Multi-granularity Reasoning Segmentation, ICLR 2025.

[2] SegLLM: Multi-round Reasoning Segmentation with Large Language Models, ICLR 2025.

**Questions:**

Please refer to the Weakness.

---

### Official Review · Reviewer_oARk · 2025-11-01

**Soundness:** 3
**Presentation:** 2
**Contribution:** 2
**Rating:** 2
**Confidence:** 4

**Summary:**

This paper introduces a modular, prompt-based system that uses a multimodal LLM (QwenVL-2.5) to generate object proposals (text labels + bounding boxes) via prompting, refines these proposals by re-prompting the LLM based on a new image with bbox proposals rendered on it and then feeds the refined boxes to SAM2 model to obtain segmentation masks. It is able to achieve high segmentation performance on segmentation tasks especially on those that require strong reasoning ability.

**Strengths:**

- This study has a practical impact as it enables general-purpose zero-shot segmentation on any image and requires no retraining.
- This pipeline system is straightforward and easy to implement. By only using a chain of prompts and off-the-shelf pretrained MLLMs, it is able to yield strong zero-shot semantic segmentation ability.
- The prompting pipeline is carefully designed, and the refining module using only previously bbox proposals and the original MLLM is interesting. The refinement module is also properly ablated to validate its necessity.

**Weaknesses:**

- Despite the interesting design of the refinement module using the generated proposals and the strong performance in zero-shot semantic segmentation tasks, the pipeline is too straightforward to have technical depth that is expected for the conference. The ability to recognize and generate bounding boxes is a by-default ability for common MLLMs since its first introduction in GPT-4. Using these bounding boxes as prompts for SAM is straightforward as well. Actually, combining MLLMs and SAM to obtain zero-shot segmentation has been a common practice in the literature, e.g. GroundedSAM. The impact of replacing GroundingDINO with MLLMs is considered marginal.
- The refinement module increases the total inference cost in a linear scale: each refinement would bring a 2x latency compared with the plain Qwen2.5VL+SAM2 baseline. Therefore, the comparison on RefCOCO and ReasonSeg is considered less fair.
- I personally think the refinement is interesting and should be analized more deeply to increase the novelty and technical impact, e.g. a deep reason why this refinement works and if more refinements bring larger performance increase.
- This one is minor as it does not influence the overal rating. More benchmarks, e.g., COCO-Stuff, Pascal VOC, and ADE20K are recommended to testify its generalization ability to enlarge its practical impact.

**Questions:**

Although the simple pipeline of integrating MLLM and SAM to achieve zero-shot segmentation and its strong performance, the study is considered not reaching the bar of novelty and technical depth for this conference.

---

### Official Review · Reviewer_SpCg · 2025-11-05

**Soundness:** 3
**Presentation:** 4
**Contribution:** 3
**Rating:** 4
**Confidence:** 3

**Summary:**

This paper introduces Seg-Agent, a completely training-free framework for language-guided segmentation. Seg-Agent guides a frozen MLLM to produce high-quality visual prompts by constructing an explicit three-step reasoning chain: (1) Generation of multiple bounding box candidates, (2) Selection of the best box from the visualized set, and (3) Refinement of the selected box's coordinates. This final, refined bounding box is then passed to a frozen foundational segmentation model, like SAM, to generate the mask. The authors also introduce a new diverse evaluation benchmark, Various-LangSeg, designed to test generalization across explicit semantic, generic object, and reasoning-guided scenarios.

**Strengths:**

1. The presentation is clear and easy to understand.
2. The method is simple, and intuitive. Results show the effectiveness.
3. The method is training-free and can be applied to various segmentation tasks.

**Weaknesses:**

1. The comparison may be unfair: The baseline makes one MLLM call. Seg-Agent makes N (augmentations, e.g., 3) + 1 (selection) + 1 (refinement) = 5 MLLM calls. This is a ~5x increase in latency and, for API-based models, a 5x increase in cost.
The paper could add experiments that compare methods of the same compute, or build baseline that use the same compute (like the best@5 random trials) to show the effectiveness of the designed method.
2. The paper does not strongly justify why this specific 3-step chain is optimal: Why are "Selection" and "Refinement" two separate steps? What if we merge them, like "Select the best box from this set and refine its coordinates". in a single call? This would cut the 5 MLLM calls down to 4. The ablation in Table 4  does not test this "combined" S+R step.
3. The Various-LangSeg benchmark is small (244 total samples). Other benchmarks like RefCOCO has several thousand validation/test images. For instance, the GOS category only has 64 samples across 4 distinct tasks (16 each).

**Questions:**

1. Since SoM is marking numbers or letters on with object mask, I want to confirm here if you use the mask or box overlaid on images for selection stage.
2. Is any experiment done about the value of k (generated candidates) vs performance?
3. The authors state that performance "heavily depends on the quality of prompt engineering". How sensitive is the model to the exact wording of the prompts in Appendix B? For example, in the refinement prompt , how much impact did the line "Note: The current bounding box may not be accurate"  have on the final result?
4. What is the result of doing simply refinement on the prediction generated on original image?

---

### Note · Authors · 2025-11-13

I have read and agree with the venue's withdrawal policy on behalf of myself and my co-authors.